# Magnetic Resonance Imaging of the Gastrointestinal Tract: Current Role, Recent Advancements and Future Prospectives

**DOI:** 10.3390/diagnostics13142410

**Published:** 2023-07-19

**Authors:** Francesca Maccioni, Ludovica Busato, Alessandra Valenti, Sara Cardaccio, Alessandro Longhi, Carlo Catalano

**Affiliations:** Department of Radiological Sciences, Pathology and Oncology, Policlinico Umberto I Hospital, Sapienza University of Rome, Viale Regina Elena 324, 00161 Rome, Italy; ludovica.busato@uniroma1.it (L.B.); alessandra.valenti@uniroma1.it (A.V.); sara.cardaccio@uniroma1.it (S.C.); alessandro.longhi@uniroma1.it (A.L.); carlo.catalano@uniroma1.it (C.C.)

**Keywords:** Magnetic Resonance Gastrointestinal Imaging, Diffusion Magnetic Resonance Imaging, Inflammatory Bowel Diseases, Gastrointestinal Neoplasms, MR Enterography, MRI Biomarkers, Intestinal Functional Imaging, Bowel Motility, Gastrointestinal Diseases, Artificial Intelligence, Contrast Media

## Abstract

This review focuses on the role of magnetic resonance imaging (MRI) in the evaluation of the gastrointestinal tract (GI MRI), analyzing the major technical advances achieved in this field, such as diffusion-weighted imaging, molecular imaging, motility studies, and artificial intelligence. Today, MRI performed with the more advanced imaging techniques allows accurate assessment of many bowel diseases, particularly inflammatory bowel disease and rectal cancer; in most of these diseases, MRI is invaluable for diagnosis, staging, and disease monitoring under treatment. Several MRI parameters are currently considered activity biomarkers for inflammation and neoplastic disease. Furthermore, in younger patients with acute or chronic GI disease, MRI can be safely used for short-term follow-up studies in many critical clinical situations because it is radiation-free. MRI assessment of functional gastro-esophageal and small bowel disorders is still in its infancy but very promising, while it is well established and widely used for dynamic assessment of anorectal and pelvic floor dysfunction; MRI motility biomarkers have also been described. There are still some limitations to GI MRI related to high cost and limited accessibility. However, technical advances are expected, such as faster sequences, more specific intestinal contrast agents, AI analysis of MRI data, and possibly increased accessibility to GI MRI studies. Clinical interest in the evaluation of bowel disease using MRI is already very high, but is expected to increase significantly in the coming years.

## 1. Introduction: GI MRI Strength and Weakness

Magnetic resonance imaging (MRI) is the most advanced cross-sectional imaging technique which currently allows the evaluation of several, if not all, intestinal segments. Compared to other imaging methods, it has several important advantages. First of all, it does not require ionizing radiation, and thus can be used in young patients, and it is repeatable and useful in the follow-up and monitoring of many benign and malignant diseases. Second, MRI has a good-to-excellent spatial resolution and an excellent tissue contrast, higher than any other imaging tool; furthermore, it is able to analyze the tissues with multiple imaging parameters, differently from any other imaging method.

MRI was firstly introduced for bowel evaluation in the late 90s [1,2], when technology and software advanced to the point where it was possible to acquire images of the entire abdomen and bowel loops in breath-hold mode, using both T1- and T2-weighted sequences, with a sufficiently high spatial resolution. One of the main challenges in the MRI evaluation of the small intestine is still related to the intrinsic motility of the bowel, i.e., intestinal peristalsis, which causes motion artifacts, together with respiratory diaphragmatic movements. This problem can be partially resolved with antispasmodic drugs but, above all, with fast imaging acquisition. Breath-holding and fast imaging are both crucial for bowel evaluation, together with a good-to-high spatial resolution, high enough to assess the thin (2 mm) normal bowel wall, and together with a wide field of view, in order to assess the entire small and large bowel. Therefore, the combination of rapid imaging acquisition, high resolution, and wide field of view are three crucial requirements for a satisfactory evaluation of the bowel, but not easy to achieve simultaneously in a single examination, even using the most advanced MRI equipment (Figure 1).

Over the past two decades, MRI techniques for bowel evaluation (here including MR Enterography, MR Enteroclysis, MR Colonography, High-Resolution MRI of the rectum, and MR Defecography) have progressively improved and today, nearly 30 years after its introduction, the clinical value of MRI in the assessment of several intestinal diseases of the small and large bowels has grown significantly. To date, relevant information can be obtained in both benign and malignant intestinal diseases in a relatively short examination time, using the multiple imaging parameters currently available, such as diffusion-weighted imaging, motility imaging, and dynamic contrast enhancement (the so-called “multiparametric MRI”) [3,4]. Recently, the diagnostic power of MRI has been added to the one offered by nuclear medicine; this advanced hybrid equipment combining MRI with PET (PET/MRI) could provide additional information on both oncological and inflammatory intestinal diseases [3,4].

GI MRI is nowadays the most accurate imaging tool for the diagnosis and monitoring of Crohn’s Disease (CD), for the preoperative evaluation of rectal cancer, and for its follow-up during treatment, as well as for the functional evaluation of pelvic floor disorders.

MRI is also able to study the motility of the esophagous, stomach, and small and large bowels, as well as of the ano-rectum, thanks to real-time dynamic fast sequences.

Some limitations, however, still exist. For example, to evaluate the gastrointestinal tract, at any site, oral contrast agents are crucial to distend the intestinal lumen and highlight the intestinal wall [5]. Disappointingly, the intestinal contrast agents currently available for GI MRI are mostly osmotic laxatives, such as polyethylene glycol (macrogol solution) or mannitol solution. Iso-osmotic laxatives, in fact, act as “biphasic contrast agents” in magnetic resonance, since their signal is hyperintense (positive) on T2-weighted images and hypointense (negative) on T1-weighted images. To distend the rectum, ultrasound gel (biphasic contrast agent) is also widely used. Unfortunately, no other specific intestinal or rectal contrast agents are currently available. Purely negative or purely positive contrast agents for magnetic resonance are no longer available, although highly “desirable” [5,6,7]. Moreover, a common cause of artifact is the presence of air, which limits the diagnostic quality of the study of the intestine, especially on 3T magnets. The sequences most frequently affected by artifacts due to residual gas include the fast spin echo and the Steady-State Free Precession-Balanced sequences (susceptibility artifacts).

Limitations and contraindications to MRI also include the presence of metallic medical implants or pacemakers, although it is not frequent, thanks to the technical evolution that has produced numerous MRI-compatible devices. Claustrophobia is rarer and rarer due to larger gantries, and it can be overcome using anxiolytic drugs, but still can be a limitation. Furthermore, GI MRI requires a longer examination time than other imaging tools, thus a high compliance of the patient to follow instruction, to stay still, and to hold breaths, which is often problematic for children or older patients, or patients with severe, painful conditions. In addition, because MRI requires a longer examination time, it is less accessible than CT and in general more expensive than other imaging techniques. Finally, longer examination time and higher costs determine a low accessibility to MRI, particularly for abdominal and gastrointestinal disorders.

The goal of this review Is to analyze the current role of MRI for the evaluation of gastrointestinal tract disorders, the pros and cons, and future perspectives.

## 2. Current Clinical Role

### 2.1. Multiparametric MRI Evaluation of Bowel Inflammation

MRI is a precious diagnostic tool in the evaluation of inflammation and inflammatory disorders. MR Enterography (MRE) is nowadays considered a fundamental exam in the evaluation ofCrohn’s disease (CD) [5], both for the initial diagnosis and for the follow-up.

To fully display the small bowel, the patient is invited to drink approximately 1500–2000 mL of iso-osmotic solution (macrogol or mannitol solutions), approximately 45 min before the examination. To reduce motion artifacts due to intestinal peristalsis, the use of an antispasmodic drug intravenously administered is suggested, before intravenous contrast injection. Distended bowel loops may be best evaluated in the prone position when possible.Basic MRE protocol for IBD includes the use of fast T2-weighted spin-echo and balanced steady-state sequences on axial and coronal planes in breath-hold or breath-hold free acquisition. Additional T2-weighted sequences may include the T2 weighted breath-hold HASTE fat-saturated thick slab in coronal acquisition (Figure 1c), or the higher resolution T2-weighted BLADE or PROPELLER sequences; furthermore, motility sequences can be useful, as well as diffusion-weighted imaging (DWI). T1-weighted dynamic contrast-enhanced MRE provides considerable contrast between the parietal lesion and the healthy wall at early (20 s), late (60 s), and delayed (5 and 7 min) phases, acquired both on axial and coronal planes. For controls during treatment, the examination can be shortened to the non-contrast sequences only.

MRE is able to recognize intestinal Crohn’s disease inflammation with several inflammatory biomarkers, some of them well known and described for over three decades, such as mural thickening, wall edema on T2 weighted images, perivisceral edema, and post-contrast enhancement on T1-weighted images [2,8,9,10,11,12] (Figure 2). Recently, additional MRI biomarkers have also been evaluated, such as restricted mural DWI signal, number, size, and signal of lymphadenopathies, stratified and delayed wall enhancement, and fibrofatty proliferation [9,10]; overall, up to 13 different MRI biomarkers have been studied in the evaluation of chronic CD inflammation, with satisfactory results [7,8].

The most recent joint guidelines of the ECCO (European Crohn’s and Colitis Organization) and ESGAR (European Society of Gastrointestinal and Abdominal Radiology) scientific societies for diagnostic assessment in IBD [10,13] state that, even though colonoscopy is the modality of choice to assess disease activity of symptomatic colonic IBD, the role of cross-sectional imaging is complementary and may be used as an alternative tool to evaluate disease activity. MRI is currently used both for the diagnosis and the follow-up, having shown a high accuracy for monitoring therapeutic responses [9]. The transmural healing, assessed by MRI, is widely considered the most reliable endpoint of the medical treatment.

Multiple CD activity scores based on MR Enterography have been proposed to stage the severity of the disease [14], such as the Magnetic Resonance Index of Activity (MaRIA), which is one of the most widely used [15], and the Magnetic Resonance Enterography Global Score (MEGS) [16]. Additional scores also include DWI, such as the Nancy and Clermont scores [11,12]. All of them are based on wall thickness and wall edema with some additional specific parameters; the Clermont is the only one which does not rely on contrast medium injection. Most of these scores require long time to be assessed. For this reason, another score has been recently proposed: the simplified MaRIA, which has a strong correlation with both the original MaRIA score and the Crohn’s Disease Endoscopic Index of Severity (CDEIS) [17]. The simplified MaRIA consists of four radiological features, which are mural thickening, mural edema, mucosal ulcerations (these three are shared with the Nancy Score), and fat stranding [17].

Due to its favorable characteristics, MRI is increasingly used in evaluation of pediatric IBD. In children affected by CD, MRI provides a safe and accurate evaluation of CD lesions in the small and large bowels, in the anorectal region, thus allowing the characterization of different CD morphological phenotypes. Furthermore, fast sequences in breath-hold free acquisition are very useful in uncooperative younger patients [18,19,20]. In the pediatric population, MRI can be used for clinical trials, being noninvasive and radiation free.

MRI is also useful in assessing most of the inflammatory disorders involving the gastrointestinal tract, not only Crohn’s Disease, such as ulcerative colitis, diverticulitis, appendicitis, and infectious colitis.

Even though colonoscopy is usually adequate to diagnose and to assess the extent and severity of ulcerative colitis, there are cases in which cross-sectional imaging, such as MRI, is fundamental, like for a difficult differential diagnosis between Crohn’s disease and ulcerative colitis or whenever endoscopy is incomplete or contraindicated [21].

In the diagnosis of diverticulitis, thanks to its superior soft tissue contrast resolution, MRI allows a very precise identification of pathological changes: the presence of pericolonic fat inflammation, thickening and inflammation of the intestinal wall, mesenteric infiltration, stenosis of colonic segments, and size of diverticula; moreover, contrast-enhanced MRI may be helpful in the differentiation of colonic diverticulitis from primary colonic carcinoma [22,23,24].

MRI can be helpful in the diagnosis of appendicitis, which can be difficult and erroneous if based on clinical-laboratory data only. MRI, providing a complete evaluation of the abdomen and pelvis, allows a correct diagnosis in up to 20% of patients with uncertain diagnosis; the most frequent differential diagnoses are adnexal disease (including ovarian torsion and hemorrhagic cysts), enterocolitis, and mesenteric adenitis, all detectable with MRI. Typical MRI features of appendicitis are a diameter greater than 7 mm, periappendiceal fat infiltration, and thickening of the appendicular wall. MRI, using a rapid, noncontrast protocol, is highly sensitive and specific in the evaluation of appendicitis, particularly in children; compared with CT, it does not use contrast or ionizing radiation and can replace it completely. However, its poor accessibility and its high cost limit its use. The fastest and least expensive MRI technique is based on a combination of T2-weighted multiplanar imaging, with and without fat suppression, and DWI. The use of intravenous gadolinium is not essential, but it increases the diagnostic accuracy [25].

Most of the imaging studies on infectious colitis focus on the role of CT. One of the most common infectious colitis types is the clostridium difficile colitis, where both CT and MRI can recognize several specific imaging parameters, such as the pancolic extension, the thickening of the intestinal wall, hyperemia, severe submucosal edema determining the typical “accordion sign”, ascites, and edema of the pericolonic fat [26,27].

Another gastrointestinal inflammatory disease recently investigated with MRI is the intestinal graft-versus-host-disease (GVHD), a severe acute or chronic complication of hematopoietic stem cell transplantation, difficult to diagnose. Multiparametric MRI can be useful for both diagnosis and staging of acute and chronic GVHD, being able to identify many specific intestinal and extraintestinal signs (biomarkers), such as predominant and continuous small bowel involvement, edema of the intestinal wall, stratified “target” wall appearance on both T2-weighted and post contrast T1-weighted images, subcutaneous fat tissue edema, and mesenteric and retroperitoneal edema [28,29].

MRI is currently considered the gold standard to diagnose and stage perianal fistulas [10,13]. MRI can accurately identify perianal fistulous tracts and abscesses, determining the involvement of both the internal and the external sphincters [30,31]. Its role is crucial for therapeutic planning and monitoring perianal fistulas during medical therapy or after surgery.

The MAGNIFI-CD score, recently developed, is an index of perianal fistula activity, and can be used as a predictive tool [32]. The MAGNIFI-CD is calculated on the following MR parameters: number of fistula tracts, hyperintensity of primary tract on post-contrast T1-weighted images, dominant feature, fistula length, extension, and inflammatory mass. The total MAGNIFI-CD score ranges from 0 to 25 [32]. The cut-off point for predicting clinical closure of a fistula is 6, with a specificity of 91% and a sensitivity of 87%. Thus, using the MAGNIFI-CD score, MRI can predict the long-term outcome of the disease.

### 2.2. MRI Evaluation of Gastrointestinal Tumours

#### 2.2.1. Gastric Cancer

In recent years, MRI has been frequently used in the diagnosis of gastric cancer, due to the development of fast, high-resolution imaging sequences that allow correct localization of parietal lesions and local staging without radiation risk. To reduce motion artifacts due to gastric peristalsis and patient breathing, the gastric cavity is distended with 500–800 cc of water with the use of an antispasmodic drug intravenously administered, similarly to CT.

Basic MRI protocols for gastric cancer include the use of fast T1- and T2-weighted spin-echo or gradient echo sequences, diffusion-weighted imaging (DWI), and intravoxel incoherent MRI (IVIM); dynamic contrast-enhanced MRI (DCE-MRI) also provides considerable contrast between the parietal lesion and the healthy wall.

Some recently developed techniques, such as the free-breathing, radial, three-dimensional (3D) sequence with star-stacked gradient echo (GRE), have a higher signal-to-noise ratio, higher contrast-to-noise ratio, and fewer artifacts, and are potentially effective in the evaluation of gastric cancer.

MRI can play a crucial role in preoperative staging, with accuracy ranging from 71.4% to 88% for T staging, and 52% to 55% for lymph node involvement; the overall accuracy of T2-weighted + DCE + DWI in T staging was significantly higher than T2-weighted + CE and T2-weighted + DWI [33,34,35,36].

The ADC value can be used for quantitative analysis as an imaging biomarker; in particular, lower ADC values were associated with higher TNM classification. Intestinal type tumors, according to Lauren’s classification, have significantly higher ADC values than diffuse type tumors, due to the presence of more distorted and narrower intercellular spaces in the diffuse type [33,34,35,36].

#### 2.2.2. Small Bowel Cancer

Small bowel tumors are rare and account for about 3% of all gastrointestinal cancers. The average interval between symptoms and diagnosis is about three years for benign tumors and about two years for malignant tumors.

The frequency of small bowel tumors decreases from proximal to distal bowel.

In addition, each histological subtype of tumor has a predilection for a different segment of the small intestine: adenocarcinoma is more frequent in the duodenum and jejunum, and carcinoid in the ileum. Small bowel tumors are generally solitary but can be multiple. Both benign and malignant tumors can cause complications, such as intussusception, obstruction, or perforation. The larger the tumor, the more likely it is to cause intestinal obstruction [37,38].

Although conventional Enteroclysis and capsule endoscopy are the most common procedures to visualize mucosal abnormalities of the small intestine, they are unable to assess the mural and extra-mural extent of small bowel neoplasms and thus to stage the tumor. Magnetic resonance Enterography has 96.6% accuracy in the diagnosis of small bowel neoplasms, it offers an excellent soft tissue contrast and multiplanar imaging, without ionizing radiations. In addition, the acquisition can be repeated over time for functional assessment of small bowel mobility, which is useful for diagnosing low-grade stenosis and determining the level of obstruction. [39]. Finally, DWI MRI can be extremely effective and sensitive in the diagnosis and characterization of intestinal lymphomas [40].

MRE, if performed with the best technical conditions (supine and prone acquisition, highly distended small bowel, spasmolytic agents, multiple T2weighted, DWI, and gadolinium-enhanced sequences) is highly accurate in the evaluation of patients with familiar intestinal polyposis, such as the Peutz-Jeghers syndrome [41]. In these patients, MRE can provide an accurate and noninvasive periodic assessment of the small bowel, with detection of polyps even smaller than 10 mm in size, thus replacing more invasive procedures (such as double balloon pushed enteroscopy), and guiding surgical or endoscopic polyp resection.

#### 2.2.3. Gastroenteropancreatic Neuroendocrine Tumors

Neuroendocrine tumors (NETs) are rare neoplasms which originate from cells of the diffuse neuroendocrine system and can occur in any body district, most frequently in the gastroenteropancreatic tract and the lung.

NETs are usually diagnosed in the metastatic phase, and the liver is the most commonly involved organ, followed by bones and lungs. Gastroenteropancreatic (GEP) NETs may exhibit unpredictable behavior ranging from indolent to highly aggressive forms [42].

Many studies have highlighted the usefulness of MRI in the study of NETs, most of them focusing on pancreatic NETs, in which the routine morphological T2-weighted and T1-weighted sequences are implemented with contrast injection and multiple post-Gadolionium scans, including arterial, venous, and delayed (>5 min), and the use of diffusion-weighted sequences. Currently, MRI is considered the best imaging technique to detect hepatic metastases; DWI sequences and T1-weighted images acquired after the injection of an hepato-biliary contrast agent (Gadolinium BOPTA or DTPA) are extremely sensitive to detect hepatic metastases, even smaller than 5 mm [43,44] (Figure 3).

Only a few studies have compared MRI to PET/CT in the diagnosis and staging of NET, with similar results: MRI was the method of choice in assessing local tumor extension and liver metastases, while PET/CT is preferred for assessing distant metastatic involvement; they concluded that the integration of the two methods provided the best results. [44,45,46,47].

#### 2.2.4. Rectal Cancer: Historical Notes & MRI T3 Stratification

Over the past 100 years, rectal cancer staging systems have undergone significant changes. One of the best-known staging classifications was developed by Sir Cuthbert Dukes in 1958, which can be traced to the current UICC/AJCC tumor-node-metastasis (TNM) staging system which classifies rectal cancers into five prognostic groups based on combinations of tumor invasion, lymph node involvement, and metastasis.

AJCC staging manuals today consider T3 tumors as a single group characterized by tumor invasion through the muscularis propria. In 1993, a subclassification of T3 tumors into four groups was proposed: T3a (<1 mm), T3b (1–5 mm), T3c (5–15 mm), and T3d (>15 mm). The degree of invasion beyond the musculature is the most sensitive way to assess tumor stage compared with lymph node status, with a significant correlation to survival [48]. High-resolution MRI has proven to be highly accurate in T3 staging, its results over-lapping histopathological measurements: MRI has shown to be equivalent to histopathology on measures of tumor spread depth [49]. Nowadays, the MRI role is crucial in the preoperative staging of rectal cancers, and in the therapeutic planning of locally advanced rectal cancer (LARC). Similarly, MRI plays a primary role in the evaluation of the therapeutic response after radio-chemotherapy of LARC, thanks to the high accuracy of morphological criteria associated with DWI [48].

In a recent 2022 article, Lambregts et al. [50] provide recommendations on how to handle current controversies in TNM-based staging of rectal cancer with MRI. To avoid inconsistencies between radiologic and pathologic reports, in tumors involving the anal canal, involvement extending to the external anal sphincter, puborectum, or levator muscles ani (i.e., skeletal muscles) should be classified as cT4b, for which a high-resolution T2-weighted coronal sequence parallel to the anal canal is required. They also propose, in accordance with the consensus guidelines on rectal MR published by the European Society of Gastrointestinal and Abdominal Radiology (ESGAR), a margin ≤ 1 mm to define an involved MRF (Meso Rectal Fascia). MRF involvement can be caused by the primary tumor or by lymph nodes, deposits, or EMVI, and it should be included in the radiology report results, with a suffix specifying whether the invasion is caused by the primary tumor or other structures, e.g., “MRF+ (primary)” or “MRF+ (non-primary).” MRF involvement should be classified as cT3 MRF+ and not cT4a.

If MRI is undoubtedly the method of choice for loco-regional staging of rectal cancer, to date its role in the local staging of colonic cancer is instead poorly investigated, since MRI of the colon is impaired by peristalsis and motion artifacts. However, 3 Tesla MRI has shown a significantly higher sensitivity compared with CT in discriminating between pT1-T3b and pT3c-T4 colon tumors. Low primary tumor ADC values have been shown to predict the risk of lymph node and/or distant metastases. Recent studies also showed that whole body MRI (WB) has similar accuracy to CT in local and distant staging. [48]. Therefore, we also expect an increasing use of MRI in staging colonic cancer; a higher MRI accuracy in local staging will provide advancements in the management of colonic cancer, similarly to rectal cancer (Figure 4).

### 2.3. MRI Evaluation of Functional Disorders

Pelvic floor dysfunctions are responsible for many pelvic disorders and symptoms, not only anorectal dysfunctions, but also urological and gynecological discomforts. For a long time, such disorders were investigated using evacuation proctography, or defecography, which is a fluoroscopic radiological technique capable of studying the pelvic floor prolapse, thanks to the opacification of the rectum and optionally also of the vagina and the small bowel. Defecography allows the real time visualization of ano-rectal movements during straining and defecation, and quantification of anorectal angles.

Nowadays, MR defecography allows a real-time visualization of the same parameters, by using a rectal contrast agent (usually ultrasound gel) to distend and visualize the rectum; dynamic images are taken during straining, squeezing, and defecation, similarly to conventional defecography [51,52,53] (Figure 5).

MRI defecography offers additional advantages compared to conventional defecography: it can carefully measure the prolapse and anorectal angles without the need for radiation exposure and allows an accurate evaluation of all pelvic floor organs, muscles, and soft tissues; moreover, functional manouvres can be repeated several times, since the patient is not exposed to radiations. The main MRI disadvantage is the obliged supine position of the patient, which is required required by most high field (1.5–3 Tesla) MRI equipment. Therefore, MR defecography evaluates pelvic floor movements in a non-physiological position, differently from conventional defecography. However, several early and more recent studies provided a good correlation between findings observed in supine and sitting positions, justifying the increasing use of MRI in the evaluation of pelvic floor disorders [54,55]. Nowadays MRI has almost completely substituted conventional defecography, due to obvious advantages: the higher spatial resolution in the static and dynamic evaluation of pelvic floor organs and of muscular structures, as well as the lack of radiation exposure.

## 3. Recent Advancements

### 3.1. Motility Imaging

The gastrointestinal tract is a moving organ whose peristaltic movement is critical to facilitate digestion and progression of food along the gastrointestinal tract, expelling waste and maximizing the action of digestive acids and enzymes on nutrients. Abnormalities of intestinal transit are frequent causes of functional disorders and abdominal pain, not associated with organic lesions. Imaging techniques that rely on static images and diagnostic tools may miss these functional alterations.

MRI, thanks to fast and cine sequences, is capable of studying the motion of the bowel and internal organs. Cine-MRI is currently widely used to study the heart; the dynamic study of the gastrointestinal tract, however, offers more difficulties than the heart, since it is wider, and bowel movements, although slower, are not periodic [56,57].

The radiological study of the esophageal movement has been for a long time a prerogative of conventional radiography, with fluoroscopy and barium swallow, but lately a few studies on the dynamic MRI assessment of esophageal dysfunctions, such as achalasia, have been published. The main limitation is the position of the patient, since in most MRI equipment the patient can perform the examination only in a supine position, while in fluoroscopy the patient can swallow the barium while standing, in a physiological position. A recent study by Biggermann et al. [58] compared MRI investigation of esophageal motility to high-resolution manometry, still considered the gold standard for assessing esophageal motor function; several MRI markers were found useful for the diagnosis of esophageal motility disorders, such as the sphincter length, the esophageal diameter (both numerically larger in patients with motility disorders), and the incomplete esophageal clearance (markers of dysmotility).

Manometry is the gold standard for the evaluation of gastric motility, but it is a long and invasive study, causing discomfort to the patient; moreover, it can quantify only the motility of the gastric fundus. Another important parameter in the study of gastric function is gastric emptying, which is often studied with nuclear medicine (gastric scintigraphy). Fluoroscopy with barium swallow is still used to study gastric motility, like for the esophagus, being easy and accessible.

Until now, gastric motility has not been extensively studied with MRI; there exist only a few articles focusing on it. The study by Heissam et al. [59], based on fifteen healthy adults, compared the motility cine-MRI, performed with semi-automated techniques, with simultaneous water perfused manometry, using water as a contrast agent and demonstrating a strong correlation between the two methods. Another study, by Hosseini et al. [60] performed cine-MRI on four healthy volunteers, using pineapple juice as contrast and demonstrating the possibility of using MRI to measure and quantify gastric motility in human participants, even quantifying different motility patterns in different gastric portions. Interestingly, both studies used natural contrast agents.

Small bowel motility is often studied in Crohn’s disease patients, as reduced motility (fixity) of inflamed loops has long been considered a hallmark of the disease [61]. To correctly assess motility, the use of an oral contrast medium to distend the intestinal loops is mandatory. A recent study by Dreja et al. [62] focused on the alteration of small bowel motility in patients with active Crohn’s disease, as compared with healthy controls. A novelty of this study was the measurement of motility not only at the level of the terminal ileum, where most of the studies are focused, but also in the jejunum and in the remaining ileum. From this study it emerged that motility alterations were localized at the level of the inflamed bowel loops, whereas distant intestinal segments were not affected. In normal subjects, the motility pattern found at the level of the terminal ileum is also observed at the level of the entire ileum, whereas the jejunum shows a different pattern.

The analysis of small bowel movements could also be crucial in the investigation of functional diseases, such as chronic pseudo-obstruction of the intestine; thus far, only a few studies have focused on the potential diagnostic role of cine-MRI in this pathology [63,64]. One of the most recent studies, conducted by Sato et al. [64], compared seven patients with chronic intestinal pseudo-obstruction and 11 healthy controls, finding that all patients had altered small bowel motility and that the severity of the disease was reflected in the severity of the cine-MRI findings.

The evaluation of colonic motility can be crucial in patients with functional constipation, a very common problem which should be accurately investigated when standard therapies fail. In clinical practice, the radiological study of colonic transit time using radiopaque markers is widely used, although it involves a moderate radiation exposure. Other radiological exams focus on the colon, such as virtual colonoscopy and colon double contrast enema, but provide only static information.

The current gold standard exams to investigate functional neuromuscular colonic disorders are colonic manometry and barostat, which are both invasive, uncomfortable for the patient, and costly, requiring anesthesia or sedation and long examination times. On the other hand, MRI is not invasive, more comfortable, and requires a shorter examination time and no anesthesia or sedation, unless in the case of claustrophobia, and also has lower costs.

One of the most important studies focused on cine-MRI of colon motility, by Kirchhoff et al. [65], evaluated ten healthy adult volunteers simultaneously performing cine-MRI and colonic manometry. This study analyzed high-amplitude propagating contractions, which are the are colonic motor patterns responsible for movements of bowel contents in anterograde direction, associated with defecation, with both manometry and cine-MRI. In this study, both of these exams were simultaneously conducted, proving that all high-amplitude propagating contractions found with manometry were also identified by cine-MRI. Cine-MRI has shown several advantages compared to manometry: it has a larger field of view and can also explore colonic segments that cannot be reached by the probe, and results do not depend on the position of the probe, which can be subject to dislocation, thus distorting the final results. On the other hand, MRI limitations include the possibility of missing the colonic movements because they are out of the examination plane of field of view.

Another recent study, by Vriesman et al. [66], was less enthusiastic. This study assessed colon motility in six pediatric patients, all of whom had symptoms and half of whom had a colostomy, with functional constipation. Cine-MRI colonic manometry was used for the identification of high-amplitude propagating contractions in response to stimulants as in the previous study, but agreement between the two methods was lower.

### 3.2. DWI and IVIM GI Imaging

Diffusion-weighted magnetic resonance imaging (DWI) provides qualitative and quantitative data on tissue cellularity based on the random diffusion of water molecules. MR-DWI improves tumour detection by providing a quantitative assessment through measurement of the apparent diffusion coefficient (ADC).

DWI has an established diagnostic value in the diagnosis, staging, and follow-up of rectal cancer [67].

The most commonly used DWI b-values for rectal cancer detection are 800 and 1000 s/mm^2^. Lower b-values are always associated with T2-shinethrough effects. DWI with ultra-high b-values (above 1000 s/mm^2^) may allow better visualization of rectal tumours due to highly effective suppression of the background signal.

Advancements in DWI technology now allow more information to be obtained for lesion detection and characterization by performing DWI with ultra-high b-values or multiple b-values. The use of DWI with a b-value of 2000 s/mm^2^ in rectal cancer helps to assess the primary rectal tumour and its response to chemoradiation therapy (CRT). [33]. However, the resolution of ultra-high b-value images is limited for clinical diagnosis. One solution is to combine high and ultra-high b-values to balance spatial resolution and functional information. Another study of patients with gastric cancer showed a high accuracy of 96.9% for tumour detection with WB-DWI/MRI. WB-DWI/MRI was highly accurate in predicting inoperability (PPV 100%, NPV 100%) compared to CT (PPV 100%, NPV 53.3%) [33]. In addition, WB-DWI/MRI revealed small peritoneal implants on the surface of the pancreas that were not detectable by laparoscopy, suggesting a higher sensitivity of WB-DWI/MRI compared to laparoscopy. According to the recent literature, the role of MRI in gastric cancer imaging has become more important with DWI and calculation of ADC as a possible biomarker in diagnosis, T-staging, and treatment response assessment. Several studies suggest that the diagnostic performance of (DWI)/MRI is not significantly different from 18F-FDG PET/CT or CT [68].

DWI is useful in assessing inflammation of the bowel wall and is widely used in the study of IBD, particularly Crohn’s disease, to diagnose the presence of intestinal inflammation associated with fibrosis, and to assess disease severity. Some of the most commonly used disease grading scores are based on DWI, such as the Nancy Score and the Clermont Score [11,12].

Intravoxel incoherent motion (IVIM) imaging has been used to estimate tissue perfusion, as blood flow in randomly oriented capillaries mimics a pseudo diffusion process. IVIM-based perfusion MRI, which does not require contrast agents, has recently gained momentum, particularly in the oncology field. In addition, IVIM allows the separation and evaluation of the contributions of perfusion and true molecular diffusion in the form of true diffusion coefficient (D slow), pseudo diffusion coefficient (D fast), and perfusion fraction (f) by using multiple b-values according to a bi-exponential model.

In oncology, the multiple b-value diffusion-driven IVIM method has already been shown to be a reliable tool for the differential diagnosis of malignant and benign tumours, as well as a promising imaging biomarker for the prognosis and treatment monitoring of various malignant tumours [10].

In GI imaging using the IVIM technique, a recent study demonstrated that there are significant differences between the whole-volume IVIM parameters of different T or N stages of gastric cancer. This study provided new quantitative prognostic tools for gastric cancer and a new imaging method for gastric cancer staging [69].

A recent study by Yoo et al. [35] also found a significant potential value of ADC in differentiating gastric stromal tumors from non-stromal tumors and in differentiating high-risk from low-risk gastric stromal tumors.

### 3.3. Hybrid Imaging: PET-MRI

Positron emission tomography (PET)/computed tomography (CT) with 2-deoxy-2-[18F]fluoro-d-glucose (FDG) is widely used for initial staging and treatment response assessment of numerous gastrointestinal malignancies. Hybrid PET/MRI scanners, which acquire PET data and MRI data simultaneously, have the potential to provide accurate whole-body staging in a single examination. Furthermore, in addition to FDG, many new PET tracers have been developed to evaluate specific aspects of tumor biology and inflammatory disorders [70,71]. Crohn’s disease is a chronic relapsing disease characterized by mucosal inflammation, lymphocytes infiltration, and fibrotic strictures. Assessment of location, extension, inflammatory activity, and severity of intestinal lesions is complex, requiring the association of endoscopic and imaging methods with histological and biochemical investigations. Nuclear medicine techniques, in particular hybrid and molecular imaging, might offer valid options in the overall assessment of disease activity. Furthermore, specific advances in nuclear medicine techniques are potentially able to assess the metabolic disease activity and to explore inflammatory pathways, providing a rational guide for the newest biological treatments [71].

### 3.4. Acquisition, Analysis, and Post Processing

Artificial intelligence (AI) is a revolutionary and still-unexplored diagnostic tool. Its application in the medical field, and in particular in radiological imaging, is becoming increasingly popular, allowing, for example, image classification, reconstruction, and resolution enhancement [72,73]. The use of AI in MRI is becoming increasingly popular, as it can reduce image acquisition times by providing very high-resolution images [74].

Two of the most developed techniques that helped to speed up image acquisition are parallel imaging and compressed sensing. With these techniques, it is possible to collect basic information from multiple coils and then reconstruct images from a smaller sample of data [74]. It is therefore likely that AI will improve the MRI assessment of normal and abnormal bowels by improving the spatial and temporal resolutions.

Many new AI tools can recognise complex patterns in data, information, and images, allowing both qualitative and quantitative assessments of radiological features.

Among the most promising methods are deep learning methods, which are able to recognise undersampled data and also allow the conversion of low-resolution data into high-resolution data [75]. Preliminary studies show the potential of a variational network to classify many different anatomical regions and achieve the diagnostic accuracy of conventional methods [76].

Magnetic resonance enterography (MRE) is an essential companion to endoscopy for the diagnosis and longitudinal monitoring of CD disease phenotype, complications, and disease activity. Crohn’s disease activity is associated with increased mural thickness and T2 signal intensity, but also with increased contrast-enhanced mural signal intensity, reflecting underlying angiogenesis within the bowel wall. Examining this signal beyond simple mean intensity may therefore provide new insights into the underlying vasculopathy in CD [77,78,79].

The texture analysis (TA) is a post-processing technique that can be applied to cross-sectional data to facilitate the analysis of heterogeneity within selected image regions. The derived image series contains features highlighted at different spatial scales, ranging from fine to coarse textures. Histogram quantification then generates the following parameters: mean of pixels within the ROI, standard deviation, symmetry of distribution, mean of positive pixels, kurtosis (distribution “accuracy” or “sharpness”), and entropy (with increasing irregularity or complexity indicated by a higher entropy value).

Artificial intelligence (AI) is set to transform the practice of medicine and the management of inflammatory bowel disease (IBD) by replicating the judgement of expert clinicians and uncovering powerful insights by analyzing volumes of data too large and complex for humans to perceive. Intelligence requires the ability to acquire, store, and logically organize information.

The emergence of artificial intelligence in IBD has been made possible by the availability of large volumes of digitized medical data and the computational methods required to analyze complex data analysis models, collectively referred to as machine learning (ML). In ML, input data is annotated, labelled, or classified, and can be clinical outcomes, expert measurements, or even physiological processes. ML methods quantify the relationships between the input data and the outcome as a model; this process is called training.

Recently, it has been possible to apply AI to improve MRI diagnoses of tumours such as rectal, breast, or even prostate cancer. Specifically for rectal cancer, we can mention the application of AI models based on “faster R-CNN” (Region-based Convolutional Neural Networks) to detect metastatic lymph nodes in the pelvises of rectal cancer patients. The study shows how multiparametric MRI (mpMRI) combined with the Mask R-CNN data processing system was able to correctly identify 80% of metastatic lymph nodes in patients with rectal cancer. The AI detection system can make the detection of lymph nodes >3 mm more efficient [76].

Other promising AI MRI models have been developed to improve the loco-regional assessment of advanced rectal cancer. These models aim to improve the accuracy of MRI in loco-regional staging, detection of metastatic lymph nodes, and assessment of response to neoadjuvant chemotherapy. With continued improvements in deep learning algorithms and the combination of MRI with deep learning image recognition, it will be possible to more accurately predict patient prognosis and response to therapy. Recently, the texture analysis of contrast-enhanced T1-weighted images was correlated to the presence or absence of histological parameters of hypoxia or angiogenesis in Crohn’s disease lesions [76]. Recent studies evaluate the use of MRI delta texture analysis (D-TA) as a methodological item able to predict the frequency of complete pathological responses and, consequently, the outcome of patients with locally advanced rectal cancer addressed to neoadjuvant chemoradiotherapy (C-RT), and subsequently, to radical surgery with good preliminary results [80,81].

In the near future, all these developing models and algorithms will certainly make AI-MRI crucial for the diagnostic approach and in-process evaluation of several GI diseases, not only rectal cancer. AI systems are proving capable of reproducing complex measurements and judgements, showing the promise of accurate and less biased measurement of disease, predicting future clinical outcomes, and enabling us to discover new insights into the pathophysiology of disease.

## 4. Future Decisions

Recent advances in MRI technology open up many possibilities for the future. It is not unreasonable to expect new techniques that may revolutionize the study of the gastrointestinal tract, such as the possibility of “magnetic resonance colonography” or “follow through MRI for dynamic evaluation of small bowel dysmotility”. The recent development of ‘Upright High Field MRI’, which allows the patient to stand or sit, may facilitate the study of gastrointestinal motility as it is performed in a physiological position. As MRI is radiation-free, new applications of this method in various bowel diseases are being sought, particularly in pediatric settings and younger adults. According to preliminary studies, magnetic resonance transfer imaging is promising in detecting and distinguishing different degrees of intestinal fibrosis in CD [82,83].

The expanded use of artificial intelligence in the evaluation of the gastrointestinal tract has not yet been as fully developed as in other systems and organs, but it seems likely to expect a real revolution in a few years.

## 5. Conclusions

MRI has a high diagnostic power in the evaluation of the bowel, from the oesophagus to the anorectum, in the assessment of functional, inflammatory, and neoplastic intestinal disorders.

In the evaluation of most functional and inflammatory gastrointestinal diseases, especially in young patients, MRI should be considered the primary diagnostic tool, being highly accurate and radiation free.

In recent years, we have witnessed the evolution of MRI techniques, which are now multiparametric and capable of providing different tissue information. The development of artificial intelligence is the new frontier that is constantly evolving to improve the diagnostic power of MRI, with ever-increasing levels of sensitivity and specificity.

Of course, there are still some limitations to the use of MRI: firstly, it requires longer examination times than CT; secondly, its cost is higher than any other imaging techniques, making it less available.

However, whenever possible, it should be considered as a reliable and primary diagnostic tool for the evaluation of most gastrointestinal diseases, as it provides an in-depth multiparametric assessment of morphological, pathological, or functional changes in both inflammatory and neoplastic diseases like no other diagnostic modality.

## Figures and Tables

**Figure 1 diagnostics-13-02410-f001:**
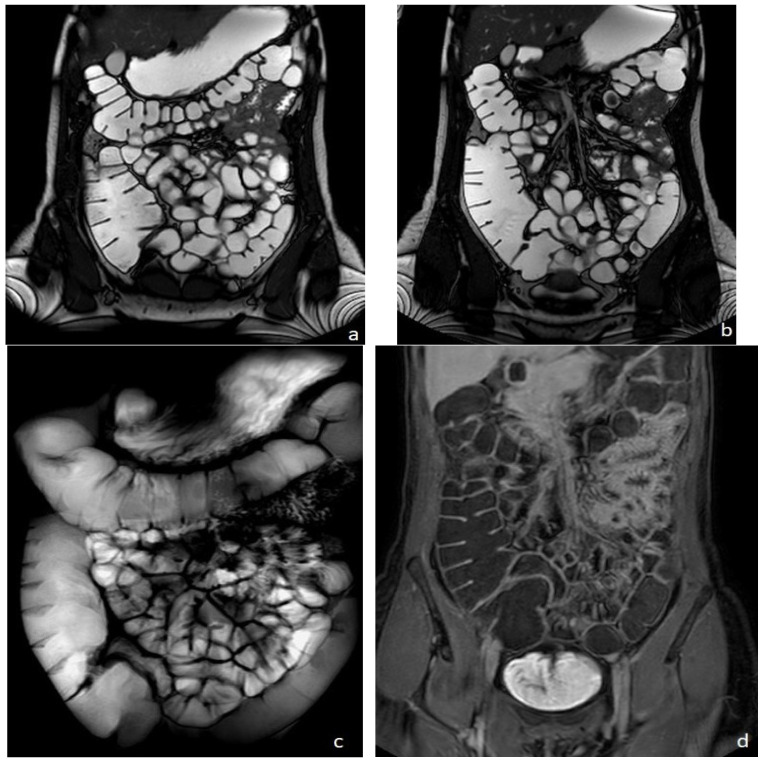
Healthy woman. MR Enterography. (**a**,**b**) are T2 balanced coronal images. (**c**) is a T2-weighted fat-suppressed single shot coronal image and (**d**) is a coronal T1-weighted image after intravenousgadolinium-chelate injection.

**Figure 2 diagnostics-13-02410-f002:**
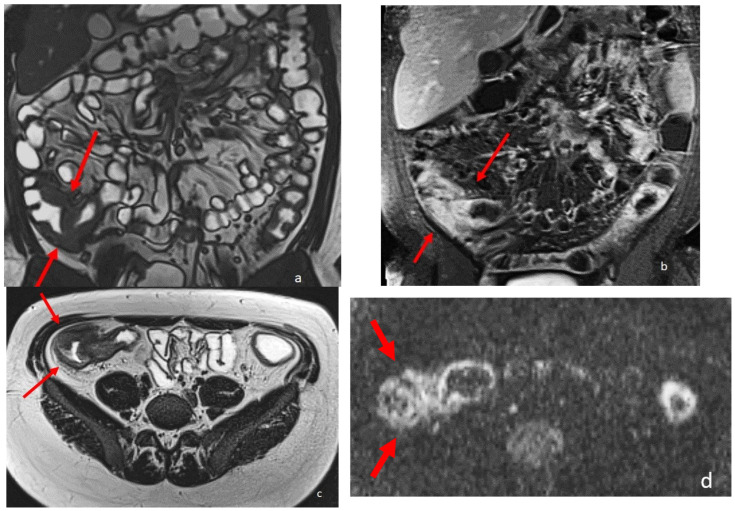
An eighteen-year-old patient with Crohn’s disease; the arrows point to the wall involvement of the last ileal loop and caecum with wall thickening, post-contrast enhancement, and restricted diffusion. (**a**) is a coronal TrueFISP image; (**b**) is a coronal contrast-enhanced T1 weighted image; and (**c**) is an AxialT2 W high-resolution BLADE image. (**d**) Axial DWI b800.

**Figure 3 diagnostics-13-02410-f003:**
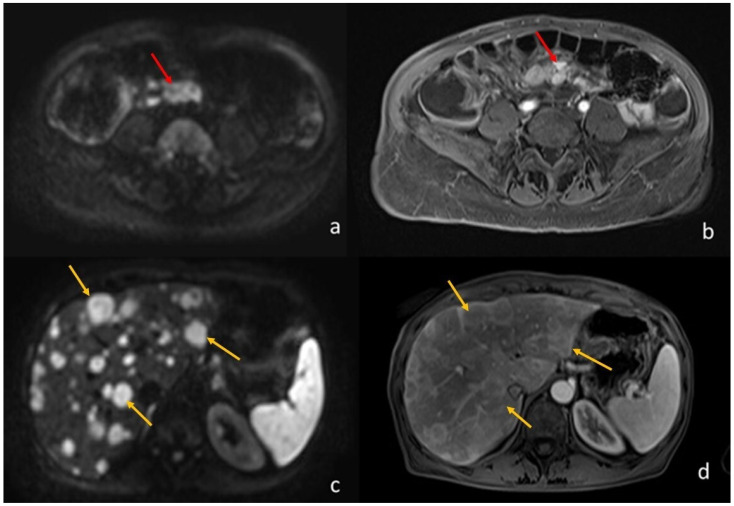
A 64-year-old woman with ileal NET and liver metastases. (**a**) (axial DWI b1000) and (**b**) (T1-weighted image post-Gadolinium injection) shows the ileal NET (pointed by the red arrows). (**c**) (axial DWI b1000) and (**d**) (T1-weighted image post-Gadolinium injection) show the multiple liver metastases (the yellow arrows point to the biggest three).

**Figure 4 diagnostics-13-02410-f004:**
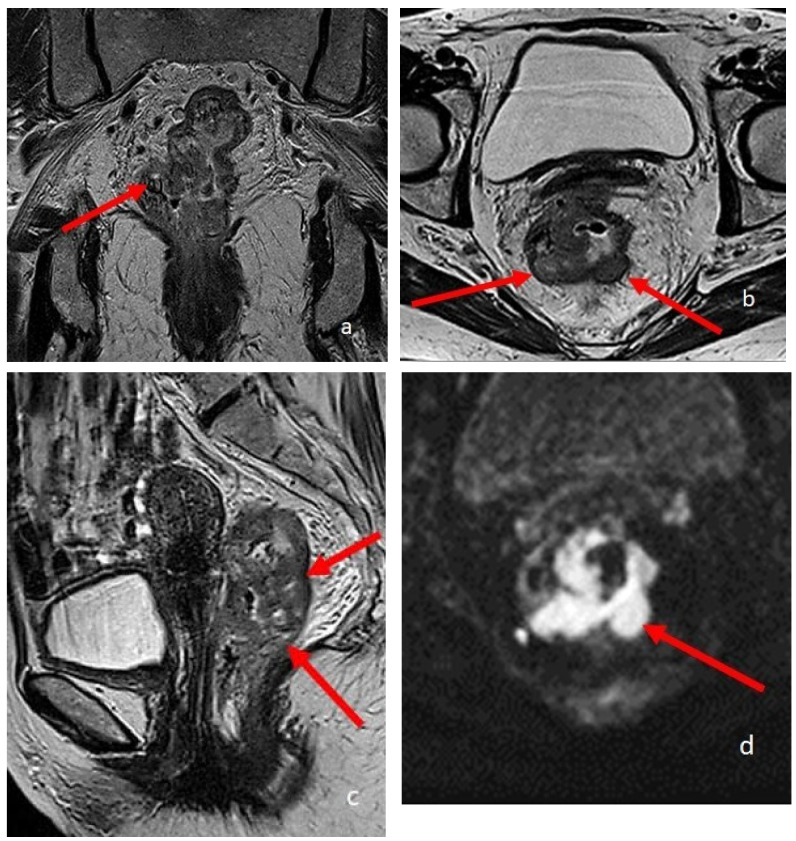
A 74-year-old patient with rectal cancer. (**a**–**c**) are T2-weighted high-resolution images in the coronal, sagittal, and axial planes, respectively. The arrows point to the pathological parietal thickening of the walls of the rectum. (**d**) is an Axial DWI b1000 image.

**Figure 5 diagnostics-13-02410-f005:**
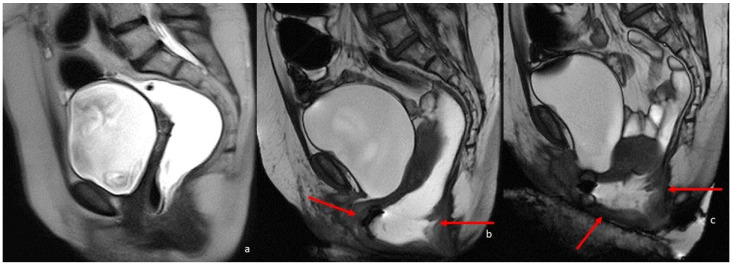
MRI defecography sagittal T2-weighted images with endoluminal US gel, in a 62-year-old female with severe stipsis. (**a**) is rest phase, (**b**) is a straining phase (the arrows indicate a severe prolapse posterior compartment with invagination and anterior rectocele; mild prolapse of middle and anterior pelvic compartments), (**c**) is in evacuation phase (showing severe three-compartment pelvic prolapse with rectal invagination and anterior rectocele cystocele and colpocele (grade III)).

## Data Availability

No new data were created for this article.

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
