# Peer review of "Magnetic Resonance Imaging of the Gastrointestinal Tract: Current Role, Recent Advancements and Future Prospectives"

_diagnostics, 2023, doi:10.3390/diagnostics13142410_

Round 1

Reviewer 1 Report

This paper presents a review of the use of MRI for diagnostics in the GI tract, including future directions.  Details of the current role of MRI in inflammatory, cancer and functional diseases are presented along with information on the newer developments including sequences and AI.  The authors also discuss limitations of the technique and where new developments may push the area forwards.

General comments.  The paper covers a topic in MRI where there has been an increase in interest over the last few years.  Although the paper touches on all aspects of GI MRI, it is not well organised and several sections repeat information given previously or include concepts which are then only expanded later in the paper, which makes it hard for the reader to follow.  Perhaps a restructuring of the sections would increase readability.  For example DWI is described in more detail in the recent advances section – however this comes after the current clinical role where several examples of its use are given but with no detail on the technique.  In addition, some scoring systems are given in detail (e.g. MAGNIFI-CD) where others are only mentioned by name (MARIA, Nancy etc). Some consistency across the sections would again help with readability. 

Specific Comments.

Introduction section.  There is no mention of gas in the lumen as a potential problem for imaging.

Lines 61-63 The references given for the statement are quite old to say recently – perhaps more recent literature reviews could be cited. 

Lines 68-69 There is no mention of colonic motility in the introduction although a paragraph on this topic is given in the recent advancements section.

The titles ‘Relevant sections’ for both sections 2 and 3 are unclear and not informative and should be removed.

Lines 111-112 appear to be a repeat of 108-110 but with extra reference.  The whole paragraph 111-118 is very similar to the one above it and the two should be merged and references updated.

Lines 209-212 Where are the references for the accuracies presented.

Line 464 – can the full name of GIST abbreviation be written out

Not sure why the molecular imaging section was included – there appear to be no papers from the GI area here and it isn’t clear that ref 65 should be included in this section.

The section on AI is not strictly all AI as texture analysis and compressed sense acquisitions are not necessarily AI based.  Please change to a more relevant title such as ‘Acquisition, analysis and post processing’

Line 550 – I don’t think this is meant to be in the paper please remove.

Conclusions normally go at the end of the paper – you could include a section on future directions prior to the conclusion.

All figures could be improved with either addition of arrows highlighting features of interest, or better descriptions of what arrows on images are point to in the figure legend.

In general the paper is written in good English.   However some minor editing would be beneficial to the overall readability.

Author Response

We thank the reviewer for valuable and accurate comments.

Regarding the suggestion to restructure the sections to increase readability, we understand your proposal, as we have already thought about the order to ensure the best understanding of the subject matter; we tried different approaches before the presentation and did not find a perfect solution, but we believe that the one used is the most balanced.

We have replaced one of the articles you pointed out with a more recent one (2018), while we have left the other one, as we think it gives an idea of the importance that MRI still had ten years ago and how it is evolving.

As for the pictures, we improved Figures 3 and 5 with arrows and improved the caption of Figure 2. We hope the result is satisfactory.  We hope the result is satisfactory.

We have corrected all the errors you have pointed out to us, and we thank you again for your valuable corrections.

We hope the result is satisfactory.

Reviewer 2 Report

 This paper titled “ Magnetic Resonance Imaging of the gastrointestinal tract: current role, recent advancements and future prospectives” provided valuable information about current role of MRI in   gastrointestinal problems. I think authors included all valuable manuscript in references. All references are well discussed. In addition, authors used all current references . 

Studies and search were conducted very well. However, I would suggest adding the Prisma graph and literature statistics as technology develops. 

I also propose to add research from the telemedicine  innovative research, i.e. analysis using on-line and network devices. 

Author Response

We thank the reviewer for valuable comments.
Unfortunately, although the Prism chart is a very useful tool for the presentation of data and statistics, we felt that the structure of our text was not suitable for the use of graphical tables such as the Prism chart, as it is a text with a very discursive structure and little numerical or statistical content. 
Unfortunately, we could not find sufficient material discussing telemedicine applied to MRI of the gastrointestinal tract.

Reviewer 3 Report

1) General comments

Dr. Maccioni, et al. reviewed “Magnetic Resonance Imaging of the gastrointestinal tract: current role, recent advancements and future prospectives”. This article is informative and well presented. The reviewer has some comments.

1.     In page 11, line 377, the authors described small bowel motility. Please describe about chronic intestinal pseudo-obstruction using cine-MRI. Please add some references, “Hidenori Ohkubo, Takaomi Kessoku, Akiko Fuyuki, et al. Assessment of small bowel motility in patients with chronic intestinal pseudo-obstruction using cine-MRI. Am J Gastroenterol 2013 Jul;108(7):1130-9. doi: 10.1038/ajg.2013.57”, and others.

1.     Please confirm the term in page 5, line 183, “clinical fallow-up”

2.     Please confirm the font in page 11, line from 372 to 373

Author Response

We thank the reviewer for valuable comments.
We have corrected the highlighted errors and included both the article you recommended and a new recent reference regarding chronic pseudo-obstruction.

Round 2

Reviewer 1 Report

Some of the comments from the original review have been addressed however there still remains some further comments to resolve.

Paragraph 106-117 is still very repetitive and now quite confusing as it gives different references for the same biomarkers.  This section needs to be rewritten and the references clearly attributed to the different biomarkers discussed.

Line 118-119 - it says references 10-13 however only 10 and 13 are ECCO guidelines.  

Something might have gone wrong with the references towards the end of the paper as ref 72 appears to be an AI paper but is referenced in the hybrid imaging section (488, 496) and at the end line 556 says refs 88-81 but should be 80-81.

The arrows in the figures should help with clarification of the anatomy if they were slightly closer to the anatomy they were highlighting.  This is particularly the case for Figure 3 and to a lesser extend figure 4.

minor corrections to the English will be necessary prior to publication

Author Response

We thank you again for your comments.

We have corrected line 118-119, we wrote [10, 13] instead of [10-13] and corrected the other references you pointed out.

We have corrected paragraph 106-117, we find it much more fluent now, we hope you agree.

We have moved the arrows in the pictures as requested.

We have also taken note of your comment on the English and edited the text again, which we now find more fluent.

We hope that it is now satisfactory.

Reviewer 3 Report

1) General comments

Dr. Maccioni, et al. revised “Magnetic Resonance Imaging of the gastrointestinal tract: current role, recent advancements and future prospectives”. This article is informative and well presented.  

Thank you for your reply. I read your responses for my questions. Your answers are precisely good, and I understood your points of view in your study. I appraise your investigation. I think that it is suitable for this journal.

Author Response

Thank you for your feedback and for your work, we are happy with your positive response